# VigDet: Knowledge Informed Neural Temporal Point Process for Coordination Detection on Social Media

**Yizhou Zhang**\*, **Karishma Sharma**\*, **Yan Liu**
Department of Computer Science
Viterbi School of Engineering
University of Southern California
{zhangyiz,krsharma,yanliu.cs}@usc.edu

## Abstract

Recent years have witnessed an increasing use of *coordinated* accounts on social media, operated by *misinformation* campaigns to influence public opinion and manipulate social outcomes. Consequently, there is an urgent need to develop an effective methodology for coordinated group detection to combat the misinformation on social media. However, the sparsity of account activities on social media limits the performance of existing deep learning based coordination detectors as they can not exploit useful prior knowledge. Instead, the detectors incorporated with prior knowledge suffer from limited expressive power and poor performance. Therefore, in this paper we propose a coordination detection framework incorporating neural temporal point process with prior knowledge such as temporal logic or pre-defined filtering functions. Specifically, when modeling the observed data from social media with neural temporal point process, we jointly learn a Gibbs distribution of group assignment based on how consistent an assignment is to (1) the account embedding space and (2) the prior knowledge. To address the challenge that the distribution is hard to be efficiently computed and sampled from, we design a theoretically guaranteed variational inference approach to learn a mean-field approximation for it. Experimental results on a real-world dataset show the effectiveness of our proposed method compared to state-of-the-art model in both unsupervised and semi-supervised settings. We further apply our model on a COVID-19 Vaccine Tweets dataset. The detection result suggests presence of suspicious coordinated efforts on spreading misinformation about COVID-19 vaccines.

## 1 Introduction

Recent research reveals that the information diffusion on social media is heavily influenced by hidden account groups [1, 30, 31], many of which are *coordinated* accounts operated by *misinformation* campaigns (an example shown in Fig. 1a). This form of abuse to spread misinformation has been seen in different fields, including politics (e.g. the election) [20] and healthcare (e.g. the ongoing COVID-19 pandemic) [31]. This persistent abuse as well as the urgency to combat misinformation prompt us to develop effective methodologies to uncover hidden coordinated groups from the diffusion cascade of information on social media.

On social media, the diffusion cascade of a piece of information (like a tweet) can be considered as a realization of a marked temporal point process where each mark of an event type corresponds to an account. Therefore, we can formulate uncovering coordinated accounts as detecting mark groups from observed point process data, which leads to a natural solution that first acquires account

---

\*Equally contributed

35th Conference on Neural Information Processing Systems (NeurIPS 2021).

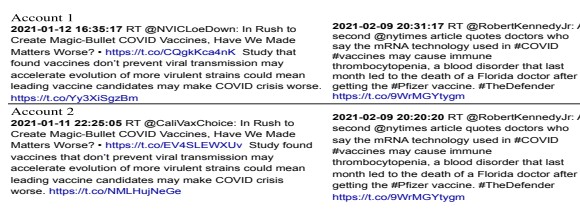
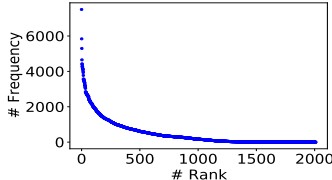

(a) Example of collaborated behaviours.                    (b) Frequency statistic of accounts.

Figure 1: The figure (a) is an example of coordinated accounts detected by our method on Twitter. They retweet similar anti-vaccine contents about COVID-19 Vaccines from same or different sources. The figure (b) is the frequency statistic of accounts in IRA dataset about the U.S. 2016 Election.

embeddings from the observed data with deep learning (e.g. neural temporal point process) and then conducts group detection in the embedding space [20, 32]. However, the data from social media has a special and important property, which is that the appearance of accounts in the diffusion cascades usually follows a long-tail distribution [18] (an example shown in Fig. 1b). This property brings a unique challenge: compared to a few dominant accounts, most accounts appear sparsely in the data, limiting the performance of deep representation learning based models. Some previous works exploiting pre-defined *collective behaviours* [2, 37, 25] can circumvent this challenge. They mainly follow the paradigm that first constructs similarity graphs from the data with some prior knowledge or hypothesis and then conducts graph based clustering. Their expressive power, however, is heavily limited as the complicated interactions are simply represented as edges with scalar weights, and they exhibit strong reliance on predefined signatures of coordination. As a result, their performances are significantly weaker than the state-of-the-art deep representation learning based model [32].

To address above challenges, we propose a knowledge informed neural temporal point process model, named Variational Inference for Group Detection (VigDet). It represents the domain knowledge of collective behaviors of coordinated accounts by defining different signatures of coordination, such as accounts that co-appear, or are synchronized in time, are more likely to be coordinated. Different from previous works that highly rely on assumed prior knowledge and cannot effectively learn from the data [2, 37], VigDet encodes prior knowledge as temporal logic and power functions so that it guides the learning of neural point process model and effectively infer coordinated behaviors. In addition, it maintains a distribution over group assignments and defines a potential score function that measures the consistency of group assignments in terms of both embedding space and prior knowledge. As a result, VigDet can make effective inferences over the constructed prior knowledge graph while jointly learning the account embeddings using neural point process.

A crucial challenge in our framework is that the group assignment distribution, which is a Gibbs distribution defined on a Conditional Random Field [17], contains a partition function as normalizer [16]. Consequently it is NP-hard to compute or sample, leading to difficulties in both learning and inference [4, 15]. To address this issue, we apply **variational inference** [22]. Specifically, we approximate the Gibbs distribution as a mean field distribution [24]. Then we jointly learn the approximation and learnable parameters with EM algorithm to maximize the evidence lower bound (ELBO) [22] of the observed data likelihood. In the E-step, we freeze the learnable parameters and infer the optimal approximation, while in the M-step, we freeze the approximation and update the parameters to maximize an objective function which is a lower bound of the ELBO with theoretical guarantee. Our experiments on a real world dataset [20] involving coordination detection validate the effectiveness of our model compared with other baseline models including the current state of the art. We further apply our method on a dataset of tweets about COVID-19 vaccine without ground-truth coordinated group label. The analysis on the detection result suggests the existence of suspicious coordinated efforts to spread misinformation and conspiracies about COVID-19 vaccines.

## 2 Related Work

### 2.1 Graph based coordinated group detection

One typical coordinated group detection paradigm is to construct a graph measuring the similarity or interaction between accounts and then conduct clustering on the graph or on the embedding acquired

by factorizing the adjacency matrix. There are two typical ways to construct the graph. One way is to measure the similarity or interaction with pre-defined features supported by prior knowledge or assumed signatures of coordinated or collective behaviors, such as co-activity, account clickstream and time sychronization [5, 29, 37]. The other way is to learn an interaction graph by fitting the data with the temporal point process models considering mutually influence between accounts as scalar scores as in traditional Hawkes Process [41]. A critical drawback of both methods is that the interaction between two accounts is simply represented as an edge with scalar weight, resulting in poor ability to capture complicated interactions. In addition, the performances of prior knowledge based methods are unsatisfactory due to reliance on the quality of prior knowledge or hypothesis of collective behaviors, which may vary with time [39].

## 2.2 Representation learning based coordinated group detection

To address the reliance to the quality of prior knowledge and the limited expressive power of graph based method, recent research tries to directly learn account representations from the observed data. In [20], Inverse Reinforcement Learning (IRL) is applied to learn the reward behind an account's observed behavior and the learnt reward is forwarded into a classifier as features. However, since different accounts' activity traces are modeled independently, it is hard for IRL to model the interactions among different accounts. The current state-of-the-art method in this direction is a neural temporal point process model named AMDN-HAGE [32]. Its backbone (AMDN), which can efficiently capture account interactions from observed activity traces, contains an account embedding layer, a history encoder and an event decoder. The account embedding vectors are optimized under the regularization of a Gaussian Mixture Model (the HAGE part). However, as a data driven deep learning model, the learning process of AMDN-HAGE lacks the guidance of prior knowledge from human. In contrast, we propose VigDet, a framework integrating neural temporal point process together and prior knowledge to address inherent sparsity of account activities.

# 3 Preliminary and Task Definition

## 3.1 Marked Temporal Point Process

A marked temporal point process (MTPP) is a stochastic process whose realization is a discrete event sequence $S = [(v_1, t_1), (v_2, t_2), (v_3, t_3), \cdots (v_n, t_n)]$ where $v_i \in \mathcal{V}$ is the type mark of event $i$ and $t_i \in \mathbb{R}^+$ is the timestamp [8]. We denote the historical event collection before time $t$ as $H_t = \{(v_i, t_i)|t_i < t\}$. Given a history $H_t$, the conditional probability that an event with mark $v \in \mathcal{V}$ happens at time $t$ is formulated as: $p_v(t|H_t) = \lambda_v(t|H_t) \exp\left(-\int_{t_{i-1}}^t \lambda_v(s|H_t)ds\right)$, where $\lambda_v(t|H_t)$, also known as intensity function, is defined as $\lambda_v(t|H_t) = \frac{\mathbb{E}[dN_v(t)|H_t]}{dt}$, i.e. the derivative of the total number of events with type mark $v$ happening before or at time $t$, denoted as $N_v(t)$. In social media data, Hawkes Process (HP) [41] is the commonly applied type of temporal point process. In Hawkes Process, the intensity function is defined as $\lambda_v(t|H_t) = \mu_v + \sum_{(v_i, t_i) \in H_t} \alpha_{v, v_i} \kappa(t - t_i)$ where $\mu_v > 0$ is the self activating intensity and $\alpha_{v, v_i} > 0$ is the mutually triggering intensity modeling mark $v_i$'s influence on $v$ and $\kappa$ is a decay kernel to model *influence decay* over time.

## 3.2 Neural Temporal Point Process

In Hawkes Process, only the $\mu$ and $\alpha$ are learnable parameters. Such weak expressive power hinders Hawkes Process from modeling complicated interactions between events. Consequently, researchers conduct meaningful trials on modeling the intensity function with neural networks [9, 21, 40, 44, 33, 23, 32]. In above works, the most recent work related to coordinated group detection is AMDN-HAGE [32], whose backbone architecture AMDN is a neural temporal point process model that encodes an event sequence $S$ with masked self-attention:

$$A = \sigma(QK^T/\sqrt{d}), \quad C = F(AV), \quad Q = XW_q, \ K = XW_k, \ V = XW_v \qquad (1)$$

where $\sigma$ is a masked activation function avoiding encoding future events into historical vectors, $X \in \mathbb{R}^{L \times d}$ ($L$ is the sequence length and $d$ is the feature dimension) is the event sequence feature, $F$ is a feedforward neural network or a RNN that summarizes historical representation from the attentive layer into context vectors $C \in \mathbb{R}^{L \times d'}$, and $W_q, W_k, W_v$ are learnable weights. Each row

$X_i$ in $X$ (the feature of event $(v_i, t_i)$) is a concatenation of learnable mark (each mark corresponds to an account on social media) embedding $E_{v_i}$, position embedding $PE_{pos=i}$ with trigonometric integral function [35] and temporal embedding $\phi(t_i - t_{i-1})$ using translation-invariant temporal kernel function [38]. After acquiring $C$, the likelihood of a sequence $S$ given mark embeddings $E$ and other parameters in AMDN, denoted as $\theta_a$, can be modeled as:

$$\log p_{\theta_a}(S|E) = \sum_{i=1}^{L} \left[ \log p(v_i|C_i) + \log p(t_i|C_i) \right],$$

$$p(v_i|C_i) = \text{softmax}(\text{MLP}(C_i))_{v_i}, \quad p(t_i|C_i) = \sum_{k=1}^{K} w_i^k \frac{1}{s_i^k \sqrt{2\pi}} \exp\left( -\frac{(\log t_i - \mu_i^k)^2}{2(s_i^k)^2} \right) \quad (2)$$

$$w_i = \sigma(V_w C_i + b_w), \quad s_i = \exp(V_s C_i + b_s), \quad \mu_i = V_\mu C_i + b_\mu$$

### 3.3 Task Definition: Coordinated Group Detection on Social Media

In coordinated group detection, we are given a temporal sequence dataset $\mathcal{S} = \{S_1, ..., S_{|D|}\}$ from social media, where each sequence $S_i = [(v_{i1}, t_{i1}), (v_{i2}, t_{i2}), \cdots]$ corresponds to a piece of information, e.g. a tweet, and each event $(v_{ij}, t_{ij})$ means that an account $v_{ij} \in \mathcal{V}$ (corresponding to a type mark in MTPP) interacts with the tweet (like comment or retweet) at time $t_{ij}$. Supposing that the $\mathcal{V}$ consists of $M$ groups, our objective is to learn a group assignment $Y = \{y_v | v \in \mathcal{V}, y_v \in \{1, ..., M\}\}$. This task can be conducted under unsupervised or semi-supervised setting. In unsupervised setting, we do not have the group identity of any account. As for the semi-supervised setting, the ground-truth group identity $Y_L$ of a small account fraction $\mathcal{V}_L \subset \mathcal{V}$ is accessible. Current state-of-the-art model on this task is AMDN-HAGE with $k$-Means. It first learns the account embeddings $E$ with AMDN-HAGE. Then it obtains group assignment Y using $k$-Means clustering on learned $E$.

## 4 Proposed Method: VigDet

In this section, we introduce our proposed model called **VigDet** (**V**ariational **I**nference for **G**roup **Det**ection), which bridges neural temporal point process and graph based method based on prior knowledge. Unlike the existing methods, in VigDet we regularize the learning process of the account embeddings with the prior knowledge based graph so that the performance can be improved. Such a method addresses the heavy reliance of deep learning model on the quality and quantity of data as well as the poor expressive power of existing graph based methods exploiting prior knowledge.

### 4.1 Prior Knowledge based Graph Construction

For the prior knowledge based graph construction, we apply co-activity [29] to measure the similarity of accounts. This method assumes that the accounts that always appear together in same sequences are more likely to be in the same group. Specifically, we construct a dense graph $\mathcal{G} = < \mathcal{V}, \mathcal{E} >$ whose node set is the account set and the weight $w_{uv}$ of an edge $(u, v)$ is the co-occurrence:

$$w_{uv} = \sum_{S \in \mathcal{S}} \mathbb{1}((u \in S) \wedge (v \in S)) \quad (3)$$

However, when integrated with our model, this edge weight is problematic because the coordinated accounts may also appear in the tweets attracting normal accounts. Although the co-occurrence of coordinated account pairs is statistically higher than other account pairs, since coordinated accounts are only a small fraction of the whole account set, our model will tend more to predict an account as normal account. Therefore, we apply one of following two strategies to acquire filtered weight $w'_{uv}$:

**Power Function** based filtering: the co-occurrence of a coordinated account pair is statistically higher than a coordinated-normal pairs. Thus, we can use a power function with exponent $p > 1$ ($p$ is a hyper-parameter) to enlarge the difference and then conduct normalization:

$$w'_{uv} = (\sum_{S \in \mathcal{S}} \mathbb{1}((u \in S) \wedge (v \in S)))^p \quad (4)$$

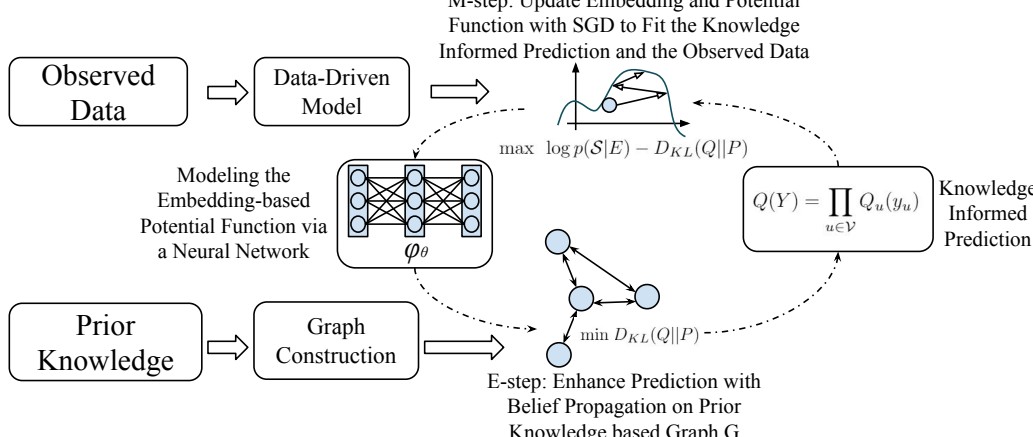

Figure 2: The overview of VigDet. In this framework, we aim at learning a knowledge informed data-driven model. To this end, based on prior knowledge we construct a graph describing the potential of account pairs to be coordinated. Then we alternately enhance the prediction of the data-driven model with the prior knowledge based graph and further update the model to fit the enhanced prediction as well as the observed data.

where $u \in S$ and $v \in S$ mean that $u$ and $v$ appear in the sequence respectively. Then the weight with relatively low value will be filtered via normalization (details in next subsection).

**Temporal Logic** [19] based filtering: We can represent some prior knowledge as a logic expression of temporal relations, denoted as $r(\cdot)$, and then only count those samples satisfying the logic expressions. Here, we assume that the active time of accounts of the same group are more likely to be similar. Therefore, we only consider the account pairs whose active time overlap is larger than a threshold (we apply half a day, i.e. 12 hours):

$$w'_{uv} = \sum_{S \in \mathcal{S}} \mathbb{1}((u \in S) \wedge (v \in S) \wedge r(u, v, S)),$$

$$r(u, v, S) = \mathbb{1}(\min(t_{ul}, t_{vl}) - \max(t_{us}, t_{vs}) > c) \tag{5}$$

where $t_{ul}, t_{vl}$ are the last time that $u$ and $v$ appears in the sequence and $t_{us}, t_{vs}$ are the first (starting) time that $u$ and $v$ appears in the sequence.

## 4.2 Integrate Prior Knowledge and Neural Temporal Point Process

To integrate prior knowledge and neural temporal point process, while maximizing the likelihood of the observed sequences $\log p(\mathcal{S}|E)$ given account embeddings, VigDet simultaneously learns a distribution over group assignments Y defined by the following potential score function given the account embeddings $E$ and the prior knowledge based graph $\mathcal{G} = <\mathcal{V}, \mathcal{E}>$:

$$\Phi(Y; E, \mathcal{G}) = \sum_{u \in \mathcal{V}} \varphi_\theta(y_u, E_u) + \sum_{(u,v) \in \mathcal{E}} \phi_\mathcal{G}(y_u, y_v, u, v) \tag{6}$$

where $\varphi_\theta(y_u, E_u)$ is a learnable function measuring how an account's group identity $y_u$ is consistent to the learnt embedding, e.g. a feedforward neural network. And $\phi_\mathcal{G}(y_u, y_v, u, v)$ is pre-defined as:

$$\phi_\mathcal{G}(y_u, y_v, u, v) = \frac{w_{uv}}{\sqrt{d_u d_v}} \mathbb{1}(y_u = y_v) \tag{7}$$

where $d_u, d_v = \sum_k w_{uk}, \sum_k w_{vk}$ are the degrees of $u, v$ and $\mathbb{1}(y_u = y_v)$ is an indicator function that equals 1 when its input is true and 0 otherwise. By encouraging co-appearing accounts to be assigned in to the same group, $\phi_\mathcal{G}(y_u, y_v, u, v)$ regularizes $E$ and $\varphi_\theta$ with prior knowledge. With the above potential score function, we can define the conditional distribution of group assignment $Y$ given embedding $E$ and the graph $\mathcal{G}$:

$$P(Y|E, \mathcal{G}) = \frac{1}{Z} \exp(\Phi(Y; E, \mathcal{G})) \tag{8}$$

where $Z = \sum_Y \exp(\Phi(Y; E, \mathcal{G}))$ is the normalizer keeping $P(Y|E, \mathcal{G})$ a distribution, also known as partition function [16, 14]. It sums up $\exp(\Phi(Y; E, \mathcal{G}))$ for all possible assignment $Y$. As a result, calculating $P(Y|E, \mathcal{G})$ accurately and finding the assignment maximizing $\Phi(Y; E, \mathcal{G})$ are both NP-hard [4, 15]. Consequently, we approximate $P(Y|E, \mathcal{G})$ with a mean field distribution $Q(Y) = \prod_{u \in \mathcal{V}} Q_u(y_u)$. To inform the learning of $E$ and $\varphi_\theta$ with the prior knowledge behind $\mathcal{G}$ we propose to jointly learn $Q$, $E$ and $\varphi_\theta$ by maximizing following objective function, which is the Evidence Lower Bound (ELBO) of the observed data likelihood $\log p(\mathcal{S}|E)$ given embedding $E$:

$$O(Q, E, \varphi_\theta; \mathcal{S}, G) = \log p(\mathcal{S}|E) - D_{KL}(Q||P) \tag{9}$$

In this objective function, the first term is the likelihood of the obeserved data given account embeddings, which can be modeled as $\sum_{S \in \mathcal{S}} \log p_{\theta_a}(S|E)$ with a neural temporal point process model like AMDN. The second term regularizes the model to learn $E$ and $\varphi_\theta$ such that $P(Y|E, \mathcal{G})$ can be approximated by its mean field approximation as precisely as possible. Intuitively, this can be achieved when the two terms in the potential score function, i.e. $\sum_{u \in \mathcal{V}} \varphi_\theta(y_u, E_u)$ and $\sum_{(u,v) \in \mathcal{E}} \phi_{\mathcal{G}}(y_u, y_v, u, v)$ agree with each other on every possible $Y$. The above lower bound can be optimized via variational EM algorithm [22, 27, 28, 34].

### 4.2.1 E-step: Inference Procedure.

In E-step, we aim at inferring the optimal $Q(Y)$ that minimizes $D_{KL}(Q||P)$. Note that the formulation of $\Phi(Y; E, \mathcal{G})$ is same as Conditional Random Fields (CRF) [17] model although their learnable parameters are different. In E-step such difference is not important as all parameters in $\Phi(Y; E, \mathcal{G})$ are frozen. As existing works about CRF [16, 14] have theoretically proven, following iterative updating function of belief propagation converges at a local optimal solution[2]:

$$Q_u(y_u = m) = \frac{\hat{Q}_u(y_u = m)}{Z_u} = \frac{1}{Z_u} \exp\{\varphi_\theta(m, E_u) + \sum_{v \in \mathcal{V}} \sum_{1 \le m' \le M} \phi_{\mathcal{G}}(m, m', u, v) Q_v(y_v = m')\} \tag{10}$$

where $Q_u(y_u = m)$ is the probability that account $u$ is assigned into group $m$ and $Z_u = \sum_{1 \le m \le M} \hat{Q}_u(y_u = m)$ is the normalizer keeping $Q_u$ as a valid distribution.

### 4.2.2 M-step: Learning Procedure.

In M-step, given fixed inference of $Q$ we aim at maximizing $O_M$:

$$O_M = \log p(\mathcal{S}|E) - D_{KL}(Q||P) = \log p(\mathcal{S}|E) + \mathbb{E}_{Y \sim Q} \log P(Y|E, \mathcal{G}) + \text{const} \tag{11}$$

The key challenge in M-step is that calculating $\mathbb{E}_{Y \sim Q} \log P(Y|E, \mathcal{G})$ is NP-hard [4, 15]. To address this challenge, we propose to alternatively optimize following theoretically justified lower bound:

**Theorem 1.** *Given a fixed inference of $Q$ and a pre-defined $\phi_{\mathcal{G}}$, we have following inequality:*

$$\mathbb{E}_{Y \sim Q} \log P(Y|E, \mathcal{G}) \ge \mathbb{E}_{Y \sim Q} \sum_{u \in \mathcal{V}} \log \frac{\exp\{\varphi_\theta(y_u, E_u)\}}{\sum_{1 \le m' \le M} \exp\{\varphi_\theta(m', E_u)\}} + const$$

$$= \sum_{u \in \mathcal{V}} \sum_{1 \le m \le M} Q_u(y_u = m) \log \frac{\exp\{\varphi_\theta(m, E_u)\}}{\sum_{1 \le m' \le M} \exp\{\varphi_\theta(m', E_u)\}} + const \tag{12}$$

The proof of this theorem is provided in the Appendix. Intuitively, the above objective function treats the $Q$ as a group assignment enhanced via label propagation on the prior knowledge based graph and encourages $E$ and $\varphi_\theta$ to correct themselves by fitting the enhanced prediction. Compared with pseudolikelihood [3] which is applied to address similar challenges in recent works [27], the proposed lower bound has a computable closed-form solution. Thus, we do not really need to sample $Y$ from $Q$ so that the noise is reduced. Also, this lower bound does not contain $\phi_{\mathcal{G}}$ explicitly in the non-constant term. Therefore, we can encourage the model to encode graph information into the embedding.

---

[2]A detailed justification is provided in the appendix

### 4.2.3 Joint Training:

The E-step and M-step form a closed loop. To create a starting point, we initialize $E$ with the embedding layer of a pre-trained neural temporal process model (in this paper we apply AMDN-HAGE) and initialize $\varphi_\theta$ via clustering learnt on $E$ (like fitting the $\varphi_\theta$ to the prediction of $k$-Means). After that we repeat E-step and M-step to optimize the model. The pseudo code of the training algorithm is presented in Alg. 1.

---
**Algorithm 1** Training Algorithm of VigDet.

---
**Require:** Dataset $\mathcal{S}$ and pre-defined $\mathcal{G}$ and $\phi_\mathcal{G}$
**Ensure:** Well trained $Q$, $E$ and $\varphi_\theta$
 1: Initialize $E$ with the embedding layer of AMDN-HAGE pre-trained on $S$.
 2: Initialize $\varphi_\theta$ on the initialized $E$.
 3: **while** not converged **do**
 4:     Acquire $Q$ by repeating Eq. 10 with $E$, $\varphi_\theta$ and $\phi_\mathcal{G}$ until convergence.{E-step}
 5:     $\varphi_\theta, E \leftarrow \mathrm{argmax}_{\varphi_\theta, E} \log p(\mathcal{S}|E) + \mathbb{E}_{Y \sim Q} \sum_{u \in \mathcal{V}} \log \frac{\exp\{\varphi_\theta(y_u, E_u)\}}{\sum_{1 \le m' \le M} \exp\{\varphi_\theta(m', E_u)\}}$. {M-step}
 6: **end while**

---

### 4.3 Semi-supervised extension

The above framework does not make use of the ground-truth label in the training procedure. In semi-supervised setting, we actually have the group identity $Y_L$ of a small account fraction $\mathcal{V}_L \subset \mathcal{V}$. Under this setting, we can naturally extend the framework via following modification to Alg. 1: For account $u \in \mathcal{V}_L$, we set $Q_u$ as a one-hot distribution, where $Q_u(y_u = y_u') = 1$ for the groundtruth identity $y_u'$ and $Q_u(y_u = m) = 0$ for other $m \in \{1, ..., M\}$.

## 5 Experiments

### 5.1 Coordination Detection on IRA Dataset

We utilize Twitter dataset containing coordinated accounts from Russia's Internet Research Agency (IRA dataset [20, 32]) attempting to manipulate the U.S. 2016 Election. The dataset contains tweet sequences (i.e., tweet with account interactions like comments, replies or retweets) constructed from the tweets related to the U.S. 2016 Election.

This dataset contains activities involving 2025 Twitter accounts. Among the 2025 accounts, 312 are identified through U.S. Congress investigations[3] as coordinated accounts and other 1713 accounts are normal accounts joining in discussion about the Election during the period of activity of those coordinated accounts. This dataset is applied for evaluation of coordination detection models in recent works [20, 32]. In this paper, we apply two settings: unsupervised setting and semi-supervised setting. For unsupervised setting, the model does not use any ground-truth account labels in training (but for hyperparameter selection, we hold out 100 randomly sampled accounts as validation set, and evaluate with reported metrics on the remaining 1925 accounts as test set). For the semi-supervised setting, we similarly hold out 100 accounts for hyperparameter selection as validation set, and another 100 accounts with labels revealed in training set for semi-supervised training). The evaluation is reported on the remaining test set of 1825 accounts. The hyper parameters of the backbone of VigDet (AMDN) follow the original paper [32]. Other implementation details are in the Appendix.

### 5.1.1 Evaluation Metrics and Baselines

In this experiment, we mainly evaluate the performance of two version of VigDet: VigDet (PF) and VigDet (TL). VigDet (PF) applies Power Function based filtering and VigDet (TL) applies Temporal Logic based filtering. For the $p$ in VigDet (PF), we apply 3. We compare them against existing approaches that utilize account activities to identify coordinated accounts.

---
[3]https://www.recode.net/2017/11/2/16598312/russia-twittertrump-twitter-deactivated-handle-list

Table 1: Results on unsupervised coordination detection (IRA) on Twitter in 2016 U.S. Election

| Method (Unsupervised) | AP | AUC | F1 | Prec | Rec | MaxF1 | MacroF1 |
|---|---|---|---|---|---|---|---|
| Co-activity | 16.9 | 52.5 | 24.6 | 17.8 | 40.7 | 27.1 | 49.5 |
| Clickstream | 16.5 | 53.2 | 21.0 | 20.6 | 21.6 | 21.0 | 53.1 |
| IRL | 23.9 | 68.7 | 35.3 | 27.5 | 49.4 | 38.6 | 58.8 |
| HP | 29.8 | 56.7 | 44.2 | 42.1 | 46.6 | 46.0 | 66.7 |
| AMDN-HAGE | 80.5 | 89.9 | 69.6 | 94.3 | 55.5 | 75.8 | 82.7 |
| AMDN-HAGE + $k$-Means | 82.0 | 93.3 | 73.0 | 90.9 | 61.2 | 77.0 | 84.5 |
| **VigDet-PL**(TL) | 83.3 | 94.0 | 70.7 | 89.6 | 59.0 | 77.8 | 83.2 |
| **VigDet-E**(TL) | 85.5 | 94.6 | 73.1 | **95.3** | 59.4 | 79.5 | 84.6 |
| **VigDet**(TL) | 86.1 | 94.6 | 73.4 | 95.1 | 59.9 | **79.6** | 84.8 |
| **VigDet-PL**(PF) | 84.5 | 95.0 | 71.9 | 91.4 | 59.6 | 79.3 | 83.9 |
| **VigDet-E**(PF) | 85.1 | 94.3 | 73.6 | 92.7 | 61.2 | 78.8 | 84.9 |
| **VigDet**(PF) | **87.2** | **95.0** | **75.2** | 91.7 | **63.9** | 79.3 | **85.7** |

**Unsupervised Baselines:** Co-activity clustering [29] and Clickstream clustering [37] are based on pre-defined similarity graphs. HP (Hawkes Process) [41] is a learnt graph based method. IRL[20] and AMDN-HAGE[32] are two recent representation learning method.

**Semi-Supervised Baselines:** Semi-NN is semi-supervised feedforward neural network without requiring additional graph structure information. It is trained with self-training algorithm [43, 26]. Label Propagation Algorithm (LPA) [42] and Graph Neural Network (GNN) (we use the GCN [13], the most representative GNN) [13, 36, 10] are two baselines incorporated with graph structure. In LPA and GNN, for the graph structures (edge features), we use the PF and TL based prior knowledge graphs (similarly used in VigDet), as well as the graph learned by HP model as edge features. For the node features in GNN, we provide the account embeddings learned with AMDN-HAGE.

**Ablation Variants:** To verify the importance of the EM-based variational inference framework and our proposed objective function in M-step, we compare our models with two variants: **VigDet-E** and **VigDet-PL** (PL for Pseudo Likelihood). In VigDet-E, we only conduct E-step once to acquire group assignments (inferred distribution over labels) enhanced with prior knowledge, but without alternating updates using the EM loop. It is similar as some existing works conducting post-processing with CRF to enhance prediction based on the learnt representations [6, 12]. In VigDet-PL, we replace our proposed objective function with pseudo likelihood function from existing works.

**Metrics:** We compare two kinds of metrics. One kind is threshold-free: Average Precision (AP), area under the ROC curve (AUC), and maxF1 at threshold that maximizes F1 score. The other kind need a threshold: F1, Precision, Recall, and MacroF1. For this kind, we apply 0.5 as threshold for the binary (coordinated/normal account) labels..

### 5.1.2 Results

**Table 1 and 2** provide results of model evaluation against the baselines averaged in the IRA dataset over five random seeds. As we can see, VigDet, as well as its variants, outperforms other methods on both unsupervised and semi-supervised settings, due to their ability to integrate neural temporal point process, which is the current state-of-the-art method, and prior knowledges, which are robust to data quality and quantity. It is noticeable that although GNN based methods can also integrate prior knowledge based graphs and representation learning from state-of-the-art model, our model still outperforms it by modeling and inferring the distribution over group assignments jointly guided by consistency in the embedding and prior knowledge space.

**Ablation Test:** Besides baselines, we also compare VigDet with its variants VigDet-E and VigDet-PL. As we can see, for Power Filtering strategy, compared with VigDet-E, VigDet achieves significantly better result on most of the metrics in both settings, indicating that leveraging the EM loop and proposed M-step optimization can guide the model to learn better representations for $E$ and $\varphi_\theta$. As for Temporal Logic Filtering strategy, VigDet also brings boosts, although relatively marginal. Such phenomenon suggests that the performance our M-step objective function may vary with the prior knowledge we applied. Meanwhile, the VigDet-PL performs not only worse than VigDet, but also

Table 2: Results on semi-supervised coordination detection (IRA) on Twitter in 2016 U.S. Election

| Method (Semi-Supervised) | AP | AUC | F1 | Prec | Rec | MaxF1 | MacroF1 |
|---|---|---|---|---|---|---|---|
| LPA(HP) | 63.3 | 76.8 | 68.1 | 76.2 | 61.8 | 71.6 | 81.5 |
| LPA(TL) | 69.7 | 85.9 | 62.3 | 88.5 | 48.6 | 66.1 | 78.6 |
| LPA(PF) | 71.1 | 85.3 | 60.8 | 66.5 | 56.4 | 68.3 | 77.2 |
| AMDN-HAGE + Semi-NN | 77.1 | 87.8 | 70.5 | 76.6 | 65.5 | 72.3 | 82.8 |
| AMDN-HAGE + GNN (HP) | 75.5 | 84.0 | 72.0 | 83.0 | 65.1 | 76.6 | 83.7 |
| AMDN-HAGE + GNN (PF) | 80.6 | 89.5 | 73.0 | 86.3 | 63.7 | 76.4 | 84.5 |
| AMDN-HAGE + GNN (TL) | 81.3 | 90.2 | 73.6 | 78.2 | **70.2** | 77.2 | 84.6 |
| **VigDet-PL**(TL) | 87.7 | 95.5 | 73.9 | 94.2 | 61.4 | 80.0 | 85.1 |
| **VigDet-E**(TL) | **88.1** | **95.7** | 73.4 | **94.6** | 60.4 | **80.8** | 84.8 |
| **VigDet**(TL) | 88.0 | **95.7** | 73.6 | 94.2 | 60.9 | **80.8** | 84.9 |
| **VigDet-PL**(PF) | 85.1 | 95.3 | 69.7 | 93.4 | 55.9 | 79.0 | 82.8 |
| **VigDet-E**(PF) | 87.1 | 95.2 | 74.4 | 92.8 | 62.4 | 79.7 | 85.3 |
| **VigDet**(PF) | 87.6 | 95.6 | **76.1** | 87.2 | 68.1 | 79.8 | **86.2** |

worse than VigDet-E. This phenomenon shows that the pseudolikelihood is noisy for VigDet and verifies the importance of our objective function.

## 5.2 Analysis on COVID-19 Vaccines Twitter Data

We collect tweets related to COVID-19 Vaccines using Twitter public API, which provides a 1% random sample of Tweets. The dataset contains 62k activity sequences of 31k accounts, after filtering accounts collected less than 5 times in the collected tweets, and sequences shorter than length 10. Although the data of tweets about COVID-19 Vaccine does not have ground-truth labels, we can apply VigDet to detect suspicious groups and then analyze the collective behavior of the group. The results bolster our method by mirroring observations in other existing researches [11, 7].

**Detection:** VigDet detects 8k suspicious accounts from the 31k Twitter accounts. We inspect tweets and account features of the detected suspicious group of coordinated accounts.

**Representative tweets:** We use topic mining on tweets of detected coordinated accounts and show the text contents of the top representative tweets in Table 3.

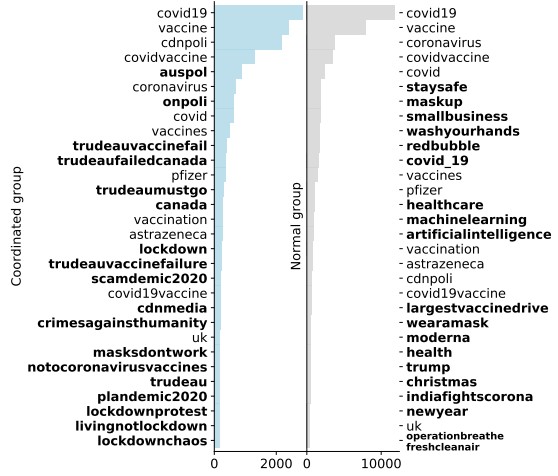

Figure 3: Top-30 hashtags in tweets of suspicious coordinated group and normal group.

**Account features:** The two groups (detected coordinated and normal accounts) are clearly distinguished in the comparison of top-30 hashtags in tweets posted by the accounts in each group (presented in Fig. 3). In bold are the non-overlapping hashtags. The coordinated accounts seem to promote that the pandemic is a hoax (#scamdemic2020, #plandemic2020), as well as anti-mask, anti-vaccine and anti-lockdown (#notcoronavirusvaccines, #masksdontwork, #livingnotlockdown) narratives, and political agendas (#trudeaumustgo). The normal accounts narratives are more general and show more positive attitudes towards vaccine, mask and prevention protocols.

Also, we measure percentage of unreliable and conspiracy news sources shared in the tweets of the detected coordinated accounts, which is 55.4%, compared to 23.2% in the normal account group. The percentage of recent accounts (created in 2020-21) is higher in coordinated group (20.4%) compared

Table 3: Representative tweets from topic clusters in tweets of detected suspicious coordinated accounts.

| |
|---|
| If mRNA vaccines can cause autoimmune problems and more severe reactions to coronavirus' maybe that's why Gates is so confident he's onto a winner when he predicts a more lethal pandemic coming down the track. The common cold could now kill millions but it will be called CV21/22? |
| This EXPERIMENTAL "rushed science" gene therapy INJECTION of an UNKNOWN substance (called a "vaccine" JUST TO AVOID LITIGATION of UNKNOWN SIDE EFFECTS) has skipped all regular animal testing and is being forced into a LIVE HUMAN TRIAL.. it seems to be little benefit to us really! |
| This Pfizer vax doesn't stop transmission,prevent infection or kill the virus, merely reduces symptoms. So why are they pushing it when self-isolation/Lockdowns /masks will still be required. Rather sinister especially when the completion date for trials, was/is 2023 |
| It is = You donf own anything, including your body. - Full and absolute ownership of your biological being. - Disruption of your immune system. - Maximizing gains for #BillGatesBioTerrorist. - #Transhumanism - #Dehumanization' |
| It is embarrassing to see Sturgeon fawning all over them. The rollout of the vaccine up here is agonisingly slow and I wouldn't be surprised if she was trying to show solidarity with the EU. There are more benefits being part of the UK than the EU. |
| It also may be time for that "boring" O'Toole (as you label him) to get a little louder and tougher. To speak up more. To contradict Trudeau on this vaccine rollout and supply mess. O'Toole has no "fire". He can't do "blood sport". He's sidelined by far right diversions. |

to 15.3% otherwise. Disinformation and suspensions are not exclusive to coordinated activities, and suspensions are based on Twitter manual process and get continually updated over time, also accounts created earlier can include recently compromised accounts; therefore, these measures cannot be considered as absolute ground-truth.

## 6 Conclusion

In this work, we proposed a prior knowledge guided neural temporal point process to detect coordinated groups on social media. Through a theoretically guaranteed variational inference framework, it integrate a data-driven neural coordination detector with prior knowledge encoded as a graph. Comparison experiments and ablation test on IRA dataset verify the effectiveness of our model and inference. Furthermore, we apply our model to uncover suspicious misinformation campaign in COVID-19 vaccine related tweet dataset. Behaviour analysis of the detected coordinated group suggests efforts to promote anti-vaccine misinformation and conspiracies on Twitter.

However, there are still drawbacks of the proposed work. First, the current framework can only support one prior knowledge based graph as input. Consequently, if there are multiple kinds of prior knowledge, we have to manually define integration methods and parameters like weight. If an automatic integration module can be proposed, we expect that the performance of VigDet can be further improved. Secondly, as a statistical learning model, although integrated with prior knowledge, VigDet may have wrong predictions, such as mislabeling normal accounts as coordinated or missing some true coordinated accounts. Therefore, we insist that VigDet should be only considered as an efficient and effective assistant tool for human verifiers or researchers to accelerate filtering of suspicious accounts for further investigation or analysis. However, the results of VigDet, including but not limited to the final output scores and the intermediate results, should not be considered as any basis or evidence for any decision, judgement or announcement.

**Acknowledgments and Disclosure of Funding**

This work is supported by NSF Research Grant CCF-1837131. Yizhou Zhang is also supported by the Annenberg Fellowship of the University of Southern California. We sincerely thank Professor Emilio Ferrara and his group for sharing the IRA dataset with us. Also, we are very thankful for the comments and suggestions from our anonymous reviewers.

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
