# VigDet: Knowledge Informed Neural Temporal Point Process for Coordination Detection on Social Media

**Yizhou Zhang***, **Karishma Sharma***, **Yan Liu**
Department of Computer Science
Viterbi School of Engineering
University of Southern California
{zhangyiz,krsharma,yanliu.cs}@usc.edu

## A    Appendix

### A.1    Proof of Theorem 1

**Theorem 1.** *Given a fixed inference of $Q$ and a pre-defined $\phi_{\mathcal{G}}$, we have following inequality:*

$$
\mathbb{E}_{Y \sim Q} \log P(Y|E, \mathcal{G}) \geq \mathbb{E}_{Y \sim Q} \sum_{u \in \mathcal{V}} \log \frac{\exp\{\varphi_\theta(y_u, E_u)\}}{\sum_{1 \leq m' \leq M} \exp\{\varphi_\theta(m', E_u)\}} + const
$$
$$
= \sum_{u \in \mathcal{V}} \sum_{1 \leq m \leq M} Q_u(y_u = m) \log \frac{\exp\{\varphi_\theta(m, E_u)\}}{\sum_{1 \leq m' \leq M} \exp\{\varphi_\theta(m', E_u)\}} + const \tag{1}
$$

*Proof.* To simplify the notation, let us apply following notations:

$$
\Phi_\theta(Y; E) = \sum_{u \in \mathcal{V}} \varphi_\theta(y_u, E_u), \qquad \Phi_{\mathcal{G}}(Y; \mathcal{G}) = \sum_{(u,v) \in \mathcal{E}} \phi_{\mathcal{G}}(y_u, y_v, u, v) \tag{2}
$$

Let us denote the set of all possible assignment as $\mathcal{Y}$, then we have:

$$
\mathbb{E}_{y \sim Q} \log P(y|E, \mathcal{G}) = \mathbb{E}_{Y \sim Q} \log \frac{\exp(\Phi(Y; E, \mathcal{G}))}{\sum_{Y' \in \mathcal{Y}} \exp(\Phi(Y'; E, \mathcal{G}))}
$$
$$
= \mathbb{E}_{y \sim Q} \Phi(Y; E, \mathcal{G}) - \log \sum_{Y' \in \mathcal{Y}} \exp(\Phi(Y'; E, \mathcal{G})) \tag{3}
$$
$$
= \mathbb{E}_{y \sim Q}(\Phi_\theta(Y; E) + \Phi_{\mathcal{G}}(Y; \mathcal{G})) - \log \sum_{Y' \in \mathcal{Y}} \exp(\Phi(Y'; E, \mathcal{G}))
$$

Because $\phi_G$ is pre-defined, $\Phi_{\mathcal{G}}(Y; \mathcal{G})$ is a constant. Thus, we have

$$
\mathbb{E}_{y \sim Q} \log P(y|E, \mathcal{G}) = \mathbb{E}_{y \sim Q} \Phi_\theta(Y; E) - \log \sum_{Y' \in \mathcal{Y}} \exp(\Phi(Y'; E, \mathcal{G})) + const \tag{4}
$$

Now, let us consider the $\log \sum_{Y' \in \mathcal{Y}} \exp(\Phi(Y'; E, \mathcal{G}))$. Since $\phi_G$ is pre-defined, there must be an assignment $Y_{\max}$ that maximize $\Phi_{\mathcal{G}}(Y; \mathcal{G})$. Thus, we have:

$$
\log \sum_{Y' \in \mathcal{Y}} \exp(\Phi(Y'; E, \mathcal{G})) \leq \log \sum_{Y' \in \mathcal{Y}} \exp(\Phi_\theta(Y; E) + \Phi_{\mathcal{G}}(Y_{\max}; \mathcal{G}))
$$
$$
= \log \exp(\Phi_{\mathcal{G}}(Y_{\max}; \mathcal{G})) \sum_{Y' \in \mathcal{Y}} \exp(\Phi_\theta(Y; E)) \tag{5}
$$
$$
= \Phi_{\mathcal{G}}(Y_{\max}; \mathcal{G}) + \log \sum_{Y' \in \mathcal{Y}} \exp(\Phi_\theta(Y; E))
$$

---

*Equally contributed

35th Conference on Neural Information Processing Systems (NeurIPS 2021), Sydney, Australia.

Since $\phi_{\mathcal{G}}$ is pre-defined, $\Phi_{\mathcal{G}}(Y_{\max}; \mathcal{G}))$ is a constant during the optimization. Note that $\sum_{Y' \in \mathcal{Y}} \exp_\theta(\Phi(Y'; E))$ sums up over all possible assignments $Y' \in \mathcal{Y}$. Thus, it is actually the expansion of following product:

$$\prod_{u \in \mathcal{V}} \sum_{1 \leq m' \leq M} \exp(\varphi_\theta(m', E_u)) = \sum_{Y' \in \mathcal{Y}} \prod_{u \in \mathcal{V}} \exp(\varphi_\theta(y'_u, E_u)) = \sum_{Y' \in \mathcal{Y}} \exp(\Phi_\theta(Y'; E)) \quad (6)$$

Therefore, for $Q$ which is a mean-field distribution and $\varphi_\theta$ which model each account's assignment independently, we have:

$$
\begin{aligned}
\mathbb{E}_{Y \sim Q} \log P(y|E, \mathcal{G}) &\geq \mathbb{E}_{y \sim Q} \Phi_\theta(Y; E) - \log \sum_{Y' \in \mathcal{Y}} \exp(\Phi_\theta(Y'; E)) + \text{const} \\
&= \mathbb{E}_{Y \sim Q} \Phi_\theta(Y; E) - \log \prod_{u \in \mathcal{V}} \sum_{1 \leq m' \leq M} \exp(\varphi_\theta(m', E_u)) + \text{const} \\
&= \mathbb{E}_{Y \sim Q} \Phi_\theta(Y; E) - \sum_{u \in \mathcal{V}} \log \sum_{1 \leq m' \leq M} \exp(\varphi_\theta(m', E_u)) + \text{const} \\
&= \mathbb{E}_{Y \sim Q} \sum_{u \in \mathcal{V}} \log \frac{\exp\{\varphi_\theta(y_u, E_u)\}}{\sum_{1 \leq m' \leq M} \exp\{\varphi_\theta(m', E_u)\}} + \text{const} \\
&= \sum_{u \in \mathcal{V}} \sum_{1 \leq m \leq M} Q_u(y_u = m) \log \frac{\exp\{\varphi_\theta(m, E_u)\}}{\sum_{1 \leq m' \leq M} \exp\{\varphi_\theta(m', E_u)\}} + \text{const}
\end{aligned}
\quad (7)
$$

$\square$

## A.2 Detailed Justification to E-step

In the E-step, to acquire a mean field approximation $Q(Y) = \prod_{u \in \mathcal{V}} Q_u(y_u)$ that minimize the KL-divergence between $Q$ and $P$, denoted as $D_{KL}(Q||P)$, we repeat following belief propagation operations until the $Q$ converges:

$$Q_u(y_u = m) = \frac{\hat{Q}_u(y_u = m)}{Z_u} = \frac{1}{Z_u} \exp\{\varphi_\theta(m, E_u) + \sum_{v \in \mathcal{V}} \sum_{1 \leq m' \leq M} \phi_{\mathcal{G}}(m, m', u, v) Q_v(y_v = m')\} \quad (8)$$

Here, we provide a detailed justification based on previous works [1, 2]. Let us recall the definition of the potential function $\Phi(Y; E, \mathcal{G})$ and the Gibbs distribution defined on it $P(Y|E, \mathcal{G})$:

$$\Phi(Y; E, \mathcal{G}) = \sum_{u \in \mathcal{V}} \varphi_\theta(y_u, E_u) + \sum_{(u,v) \in \mathcal{E}} \phi_{\mathcal{G}}(y_u, y_v, u, v) \quad (9)$$

$$P(Y|E, \mathcal{G}) = \frac{1}{Z} \exp(\Phi(Y; E, \mathcal{G})) \quad (10)$$

where $Z = \sum_Y \exp(\Phi(Y; E, \mathcal{G}))$. With above definitions, we have the following theorem:

**Theorem 2.** *(Theorem 11.2 in [1])*

$$D_{KL}(Q||P) = \log Z - \mathbb{E}_{Y \sim Q} \Phi(Y; E, \mathcal{G}) - H(Q) \quad (11)$$

*where $H(Q)$ is the information entropy of the distribution $Q$.*

A more detailed derivation of the above equation can be found in the appendix of [2]. Since $Z$ is fixed in the E-step, minimizing $D_{KL}(Q||P)$ is equivalent to maximizing $\mathbb{E}_{Y \sim Q} \Phi(Y; E, \mathcal{G}) + H(Q)$. For this objective, we have following theorem:

**Theorem 3.** *(Theorem 11.9 in [1]) $Q$ is a local maximum if and only if:*

$$Q_u(y_u = m) = \frac{1}{Z_u} \exp(\mathbb{E}_{Y - \{y_u\} \sim Q} \Phi(Y - \{y_u\}; E, \mathcal{G}|y_u = m)) \quad (12)$$

*where $Z_u$ is the normalizer and $\mathbb{E}_{Y - \{y_u\} \sim Q} \Phi(Y - \{y_u\}; E, \mathcal{G}|y_u = m)$ is the conditional expectation of $\Phi$ given that $y_u = m$ and the labels of other nodes are drawn from $Q$.*

Meanwhile, note that the expectation of all terms in $\Phi$ that do not contain $y_u$ is invariant to the value of $y_u$. Therefore, we can reduce all such terms from both numerator (the exponential function) and denominator (the normalizer $Z_u$) of $Q_u$. Thus, we have following corollary:

**Corollary 1.** *$Q$ is a local maximum if and only if:*

$$Q_u(y_u = m) = \frac{1}{Z_u} \exp\{\varphi_\theta(m, E_u) + \sum_{v \in \mathcal{V}} \sum_{1 \leq m' \leq M} \phi_\mathcal{G}(m, m', u, v) Q_v(y_v = m')\} \qquad (13)$$

*where $Z_u$ is the normalizer*

A more detailed justification of the above corollary can be found in the explanation of Corollary 11.6 in the Sec 11.5.1.3 of [1]. Since the above local maximum is a fixed point of $D_{KL}(Q\|P)$, fixed-point iteration can be applied to find such local maximum. More details such as the stationary of the fixed points can be found in the Chapter 11.5 of [1]

### A.3 Details of Experiments

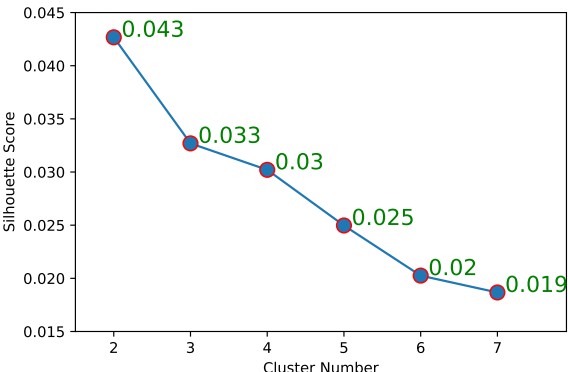

Figure 1: The silhouette scores of different group number.

#### A.3.1 Implementation details on IRA dataset

We split the sequence set to 75/15/15 fractions for training/validation/test sets. For the setting of AMDN and AMDN-HAGE [3] we use the default setting from the original paper including activity sequences of maximum length 128 (we split longer sequences), batch size of 256 (on 1 NVIDIA-2080Ti gpu), embedding dimension of 64, number of mixture components for the PDF in the AMDN part of 32, single head and single layer attention module, component number in the HAGE part of 2. Our implementation is totally based on PyTorch and Adam optimizer with 1e-3 learning rate and 1e-5 regularization (same as [3]). The number of loops in the EM algorithm is picked up from $\{1, 2, 3\}$ based on the performance on the validation account set. In each E-step, we repeat the belief propagation until convergence (within 10 iterations) to acquire the final inference. In each M-step, we train the model for max 50 epochs with early stopping based on validation objective function. The validation objective function is computed from the sequence likelihood on the 15% held-out validation sequences, and KL-divergence on the whole account set based on the inferred account embeddings in that iteration.

#### A.3.2 Implementation details on COVID-19 Vaccine Tweets dataset

We apply the Cubic Function based filtering because it shows better performance on unsupervised detection on IRA dataset. We follow all rest the settings of VigDet (CF) in IRA experiments except the GPU number (on 4 NVIDIA-2080Ti). Also, for this dataset, since we have no prior knowledge about how many groups exist, we first pre-train an AMDN by only maximizing its observed data likelihood on the dataset. Then we select the best cluster number that maximizes the silhouette score as the group number. The final group number we select is 2. The silhouette scores are shown in Fig.

Table 1: Results on unsupervised coordination detection (IRA) on Twitter in 2016 U.S. Election

| Method | AP | AUC | F1 | Prec | Rec | MaxF1 | MacroF1 |
|---|---|---|---|---|---|---|---|
| Co-activity | $.169 \pm .01$ | $.525 \pm .03$ | $.246 \pm .02$ | $.178 \pm .02$ | $.407 \pm .07$ | $.271 \pm .01$ | $.495 \pm .02$ |
| Clickstream | $.165 \pm .01$ | $.532 \pm .01$ | $.21 \pm .02$ | $.206 \pm .02$ | $.216 \pm .03$ | $.21 \pm .02$ | $.531 \pm .01$ |
| IRL | $.239 \pm .01$ | $.687 \pm .02$ | $.353 \pm .03$ | $.275 \pm .03$ | $.494 \pm .05$ | $.386 \pm .01$ | $.588 \pm .02$ |
| HP | $.298 \pm .03$ | $.567 \pm .03$ | $.442 \pm .03$ | $.421 \pm .02$ | $.466 \pm .04$ | $.46 \pm .03$ | $.667 \pm .01$ |
| A-H | $.805 \pm .03$ | $.899 \pm .02$ | $.696 \pm .05$ | $.943 \pm .03$ | $.555 \pm .06$ | $.758 \pm .03$ | $.827 \pm .03$ |
| A-H(Kmeans) | $.82 \pm .05$ | $.933 \pm .03$ | $.73 \pm .04$ | $.909 \pm .03$ | $.612 \pm .05$ | $.77 \pm .03$ | $.845 \pm .02$ |
| **VigDet-PL**(NF) | $.816 \pm .05$ | $.933 \pm .03$ | $.73 \pm .04$ | $.852 \pm .04$ | $\mathbf{.641 \pm .06}$ | $.765 \pm .05$ | $.844 \pm .02$ |
| **VigDet-E**(NF) | $.868 \pm .03$ | $\mathbf{.955 \pm .01}$ | $.692 \pm .07$ | $\mathbf{.964 \pm .03}$ | $.543 \pm .04$ | $.792 \pm .04$ | $.825 \pm .04$ |
| **VigDet**(NF) | $.856 \pm .03$ | $.951 \pm .02$ | $.698 \pm .04$ | $.958 \pm .03$ | $.551 \pm .05$ | $.788 \pm .03$ | $.828 \pm .03$ |
| **VigDet-PL**(TL) | $.833 \pm .05$ | $.94 \pm .03$ | $.707 \pm .06$ | $.896 \pm .05$ | $.59 \pm .08$ | $.778 \pm .04$ | $.832 \pm .03$ |
| **VigDet-E**(TL) | $.855 \pm .03$ | $.946 \pm .03$ | $.731 \pm .03$ | $.953 \pm .03$ | $.594 \pm .04$ | $\mathbf{.796 \pm .03}$ | $.846 \pm .02$ |
| **VigDet**(TL) | $.861 \pm .03$ | $.946 \pm .03$ | $.734 \pm .03$ | $.951 \pm .03$ | $.599 \pm .04$ | $\mathbf{.796 \pm .03}$ | $.848 \pm .02$ |
| **VigDet-PL**(CF) | $.845 \pm .04$ | $.95 \pm .02$ | $.719 \pm .05$ | $.914 \pm .04$ | $.596 \pm .07$ | $.793 \pm .03$ | $.839 \pm .03$ |
| **VigDet-E**(CF) | $.851 \pm .04$ | $.943 \pm .03$ | $.736 \pm .03$ | $.928 \pm .03$ | $.612 \pm .04$ | $.789 \pm .03$ | $.849 \pm .02$ |
| **VigDet**(CF) | $\mathbf{.872 \pm .03}$ | $.95 \pm .03$ | $\mathbf{.752 \pm .03}$ | $.917 \pm .04$ | $.639 \pm .04$ | $.793 \pm .03$ | $\mathbf{.857 \pm .02}$ |

Table 2: Results on semi-supervised coordination detection (IRA) on Twitter in 2016 U.S. Election

| Method | AP | AUC | F1 | Prec | Rec | MaxF1 | MacroF1 |
|---|---|---|---|---|---|---|---|
| LPA(HP) | $.633 \pm .09$ | $.768 \pm .04$ | $.681 \pm .05$ | $.762 \pm .06$ | $.618 \pm .06$ | $.716 \pm .05$ | $.815 \pm .03$ |
| LPA(TL) | $.697 \pm .04$ | $.859 \pm .02$ | $.623 \pm .06$ | $.885 \pm .03$ | $.486 \pm .08$ | $.661 \pm .05$ | $.786 \pm .03$ |
| LPA(CF) | $.711 \pm .04$ | $.853 \pm .02$ | $.608 \pm .04$ | $.665 \pm .03$ | $.564 \pm .07$ | $.683 \pm .06$ | $.772 \pm .02$ |
| A-H + Semi-NN | $.771 \pm .04$ | $.878 \pm .03$ | $.705 \pm .04$ | $.766 \pm .04$ | $.655 \pm .04$ | $.723 \pm .04$ | $.828 \pm .02$ |
| A-H + GNN (HP) | $.755 \pm .06$ | $.84 \pm .05$ | $.72 \pm .07$ | $.83 \pm .14$ | $.651 \pm .08$ | $.766 \pm .05$ | $.837 \pm .04$ |
| A-H + GNN (CF) | $.806 \pm .06$ | $.895 \pm .04$ | $.73 \pm .07$ | $.863 \pm .06$ | $.637 \pm .09$ | $.764 \pm .06$ | $.845 \pm .04$ |
| A-H + GNN (TL) | $.813 \pm .05$ | $.902 \pm .03$ | $.736 \pm .06$ | $.782 \pm .06$ | $.702 \pm .09$ | $.772 \pm .06$ | $.846 \pm .03$ |
| **VigDet-PL**(NF) | $.865 \pm .03$ | $.954 \pm .01$ | $.698 \pm .06$ | $.956 \pm .03$ | $.553 \pm .07$ | $.796 \pm .04$ | $.828 \pm .03$ |
| **VigDet-E**(NF) | $.868 \pm .03$ | $.955 \pm .01$ | $.692 \pm .07$ | $\mathbf{.964 \pm .03}$ | $.543 \pm .07$ | $.792 \pm .04$ | $.825 \pm .04$ |
| **VigDet**(NF) | $.871 \pm .03$ | $.956 \pm .01$ | $.712 \pm .06$ | $.944 \pm .04$ | $.575 \pm .07$ | $.795 \pm .04$ | $.836 \pm .03$ |
| **VigDet-PL**(TL) | $.877 \pm .04$ | $.955 \pm .01$ | $.739 \pm .08$ | $.942 \pm .04$ | $.614 \pm .09$ | $.80 \pm .06$ | $.851 \pm .04$ |
| **VigDet-E**(TL) | $\mathbf{.881 \pm .04}$ | $\mathbf{.957 \pm .01}$ | $.734 \pm .08$ | $.946 \pm .04$ | $.604 \pm .09$ | $\mathbf{.808 \pm .05}$ | $.848 \pm .04$ |
| **VigDet**(TL) | $.88 \pm .04$ | $\mathbf{.957 \pm .01}$ | $.736 \pm .08$ | $.942 \pm .04$ | $.609 \pm .09$ | $\mathbf{.808 \pm .05}$ | $.849 \pm .04$ |
| **VigDet-PL**(CF) | $.851 \pm .04$ | $.953 \pm .01$ | $.697 \pm .06$ | $.934 \pm .03$ | $.559 \pm .07$ | $.79 \pm .04$ | $.828 \pm .03$ |
| **VigDet-E**(CF) | $.871 \pm .04$ | $.952 \pm .01$ | $.744 \pm .06$ | $.928 \pm .03$ | $.624 \pm .08$ | $.797 \pm .05$ | $.853 \pm .04$ |
| **VigDet**(CF) | $.876 \pm .03$ | $.956 \pm .01$ | $\mathbf{.761 \pm .06}$ | $.872 \pm .07$ | $\mathbf{.681 \pm .09}$ | $.798 \pm .05$ | $\mathbf{.862 \pm .04}$ |

1. After that, we train the VigDet on the dataset with group number of 2. As for the final threshold we select for detection, we set it as 0.8 because it maximizes the silhouette score on the final learnt embedding[2].

### A.4 Detailed Performance

In Table. 1 and 2, we show detailed performance of our model and the baselines. Specifically, we provide the error bar of different methods. Also in the Sec. 4.1, we mention that we design two strategies to filter the edge weight because the naive edge weights suffer from group unbalance. Here, we give detailed results of applying naive edge weight without filtering in VigDet (denoted as VigDet (NF)). As we can see, compared with the version with filtering strategies, the recall scores of most variants with naive edge weight are significantly worse, leading to poor F1 score (excpet VigDet-PL(NF) in unsupervised setting, which performs significantly worse on threshold-free metrics like AP, AUC and MaxF1).