# OpenReview forum: "VigDet: Knowledge Informed Neural Temporal Point Process for Coordination Detection on Social Media"
_NeurIPS.cc/2021/Conference — NeurIPS 2021 Poster_

### Official Review · Reviewer_w5yn · 2021-07-09

**Rating:** 5
**Confidence:** 3

**Summary:**

This paper propose a method for coordinated group detection in social networks. For group detection they use existing neural temporal point process to acquire account embeddings from the observed data and then apply group detection in that embedding space. Moreover, to incorporate prior knowledge in their method they use co-activity matrix, which is filtered by cubic and/or temporal logic functions to better compensate for coordinated accounts in the prior. To solve the resulting inference problem, they approximate likelihood as a mean field and jointly learn the approximation and learnable parameters with EM algorithm.


**Limitations And Societal Impact:**

The authors did not mentioned the societal impact of their, although it may have critical impact.

**Main Review:**

The proposed method is incremental and only incorporate filtered co-activity matrix to neural temporal point process embedding for group clustering. The inference section (4.2) is not easy to follow, the authors should elaborate more on this section. Also, the reason/intuition behind potential function Eq.6 is not clear.


**Time Spent Reviewing:**

4

---

> ### Author Response · Authors · 2021-08-10
> **Response to Reviewer w5yn**
>
> Thank you for the reviews. We clarify and respond to the comments below:
>
> (1) Comment: The proposed method is incremental and only incorporates filtered co-activity matrix to neural temporal point process embedding for group clustering.
>
> Response: The proposed work is a novel learning framework (with variational inference) to integrate prior domain knowledge to guide data-driven learning. Any kind of data-driven model (e.g. AMDN-HAGE) and a broad type of prior knowledge can be plugged in.
> Our framework provides a more convenient, effective, and efficient way to incorporate domain knowledge and combine it with data-driven learning for coordination detection. The prior knowledge based graph can be easily customized by the nature of the domain. And the framework allows the model to rely on signals from both data and rich domain knowledge to detect coordination groups more effectively than other methods.
>
>
>
> (2) Comment: The inference section (4.2) is not easy to follow, the authors should elaborate more on this section.
>
> Response: Thank you for the feedback. The inference section infers P(Y|E, G) i.e., probability of which group each account belongs to given the embeddings and prior knowledge based graph.
>
> P(Y|E, G) is represented by the potential score function with the normalization constant (Eq 8). Since for M groups and V accounts, the normalization constant requires integrating over all M^V possible configurations of group assignments, making it intractable to compute.
>
> Therefore, we approximate it using a variational distribution Q(Y) and learn it by optimizing the evidence lower bound (ELBO) with a variational EM algorithm (where the M-step is approximated with the proposed theoretically derived lower-bound with closed-form solution).
> We will improve the writing to make it more clear to readers.
>
> (3) Comment: Also, the reason/intuition behind potential function Eq.6 is not clear.
>
> Response: The first term “is a learnable function measuring how an account’s group identity $y_u$ is consistent to the learnt embedding” (line 173-174) and the second term “By encouraging co-appearing accounts to be assigned into the same group, \phi_G regularizes $E$ and $\varphi_\theta$ with prior knowledge.” (line 176-177).
>
> (4) Comment: The authors did not mention the societal impact of their work, although it may have a critical impact.
>
> Response: Thank you for the feedback. In terms of societal impact, coordinated misinformation campaigns are critical to identify, and automated detection can accelerate human verification. But as noted in the precision-recall in Table 1 and 2, false positives can occur. For the limitations and societal impact, we discussed them in Checklist 1(b) (and in Ablation section 5.1.2).  We will include it in a separate Discussion section in the paper for clarity.

---

> ### Author Response · Authors · 2021-08-31
> **Update**
>
> Dear Reviewer w5yn,
>
> Is there still any more clarification required or more questions based on the initial responses? We are willing to provide more clarifications if so.
>
>
> Best regards,
>
> Authors of Paper4983

---

### Official Review · Reviewer_P6ya · 2021-07-12

**Rating:** 7
**Confidence:** 3

**Summary:**

The paper presents a new, neural change point process that incorporates a prior knowledge graph, for detecting coordinated accounts on social media. The paper also presents an approximate formulation of the original NP-Hard formulation of their model, which they then train by use of an E-M algorithm.

**Limitations And Societal Impact:**

Limitations and Societal impact are appropriately addressed. The work has the potential for a very positive societal impact, given the use of coordination of online accounts in disinformation campaigns.

**Main Review:**

•	Originality: The paper is of moderate originality. The paper is attacking a recent problem that is of huge significance to society, namely that of finding coordination among online accounts, which is a key element of many harmful, disinformation campaigns. The technical contribution, however, is not as novel as the problem in that it is an incremental improvement on existing algorithms (i.e. improvement over ADMN-HAGE); the proposed model incorporates a prior and has a better training procedure, but the core part of the model, AMDN is the same. Additionally, the core of the prior knowledge graph construction also exists in other works. For example, the construction of co-occurrence graphs in time has been used for finding account coordination (Pachecho et al., 2021, Giglietto, Coordinated Link Sharing Behavior as a Signal to Surface Sources of Problematic Information on Facebook, 2020, and Horawalavithana, Malicious and Low Credibility URLs on Twitter During the AstraZeneca COVID-19 Vaccine Development, 2021). That said, to the best of the reviewer's knowledge, such prior knowledge graphs have not been used in the novel way they are in the paper.

•	Quality: the paper is of good quality overall. The paper does a good job proving its theoretical claims and relating its model to good theoretical foundations (i.e. temporal point processes, Hawkes Process, and Conditional Random Fields). One question from the theoretical grounding is in section 4.1, why does the cubic weighting scheme work as a better weighting for the graph. Some intuition or sources as o why it works would be useful. The empirical validation is also compelling. The paper not only evaluates in semi-supervised and unsupervised settings on two different data sets but also does an ablation study. That said, there are a couple of areas that could be improved on empirical validation. For the sequences of tweets, is an interaction, which would appear in $S$, just interacting with a tweet, such as retweeting or quoting? And is this the same for both empirical tests (i.e. IRA data set and the COVID data set). Are the labels being evaluated with the metrics ‘normal’ and ‘coordinated’ or is a different labeling scheme used (i.e. just whether an account in the given data is participating in coordination, without respect to the nature of that coordination)? How was the prior graph for the 5.2 experiment constructed? Was it the CF version or the TL version? For the 5.2 experiment, it may be worth mentioning that the results mirror those observed in other studies on COVID (Horawalavithana Malicious and Low Credibility URLs on Twitter During the AstraZeneca COVID-19 Vaccine Development, 2021, and Cruickshank, Clustering Analysis of Website Usage on Twitter During the COVID-19 Pandemic, 2021) to bolster the results (since there are no labels for validation).

•	Clarity: The clarity is overall good. The writing is good with a few minor typos (i.e. ‘during during’ on line 231). I do wish there was some kind of overview figure for the method that details how you go from tweet data through the method to coordinated accounts. For Figure 2(b), while I do appreciate the attempt at showing the intuition, I think this figure would be more successful with including intuitive word descriptions of the figures (steps) instead of the mathematical notation.

•	Significance: The paper is of moderate significance. While it is certainly attacking a very significant problem and displays some positive empirical results there are some issues with its real-world utility. First, the model seems to be predicated upon temporal sequences that include users interacting with tweets, like retweeting. Recent work has shown that other types of coordination, like reposting of the same URLs or reposting the same tweet, but with different mentions (i.e. flooding or spamming) or reporting of the same tweet text from different accounts to also be prevalent and tied to serious disinformation campaigns. In each case, it’s not clear the proposed method would successfully find that coordination. Also, It’s not clear that this method, since it relies on interactions with the content and users of one platform, could be extended to multi-platform coordination (for examples: Horawalavithana, Twitter Is the Megaphone of Cross-platform Messaging on the White Helmets, 2020, Ng, Multi-platform Information Operations: Twitter, Facebook and YouTube against the White Helmets, 2020, Starbird, Disinformation as Collaborative Work: Surfacing the Participatory Nature of Strategic Information Operations, 2020).

------------------ Following Author's Comments --------------
The authors have clarified my main issues with the paper. Namely, why the cubic weighting scheme was used and whether the proposed approach can be used with other forms of coordination seen in disinformation spread (i.e. cross-platform, coordinated link-sharing behavior, etc.). I do think there are some areas for future work in how to characterization all of the various forms of coordination into the point process as well as having different cells or even types of coordination. However, I do believe if the clarification points are included in the manuscript (i.e. cubic weighting scheme intuition, the reworking of the methods section diagrams, etc.) that this paper should move forward as it does have an important societal impact and presents a novel combination of data-based learning and expert knowledge in its model.

**Time Spent Reviewing:**

3

---

> ### Author Response · Authors · 2021-08-10
> **Response to Reviewer P6ya**
>
> Thank you for the reviews. We clarify and answer the questions below:
>
> (1) Question: One question from the theoretical grounding is in section 4.1, why does the cubic weighting scheme work as a better weighting for the graph. Some intuition or sources as to why it works would be useful.
>
> Answer: The intuition is that the cubic weighing scheme, together with the normalization in Equation 7, filters out the low weight edges. The cubic function enlarges the difference between high weight edges and low weight edges and thus the normalization step makes those low weight edges much lighter. On social media, low weight edges between two users may be caused simply by coincidence or similar interest and high weight edges are more likely to be the cue of coordination. Therefore, the cubic function helps improve the performance.
>
>
> (2) Question: For the sequences of tweets, is an interaction, which would appear in S, just interacting with a tweet, such as retweeting or quoting? And is this the same for both empirical tests (i.e. IRA data set and the COVID data set)?
>
> Answer: The sequence of tweets is used to represent the activity traces of accounts. It constitutes a posted tweet (or content) with subsequent chains of account engagements. Engagements can be in the form of replies/retweets/quotes on the posted tweet. As well as subsequent retweets/replies/quotes of earlier replies/retweets/quotes in the sequence.
>
>
> (3) Question: Are the labels being evaluated with the metrics ‘normal’ and ‘coordinated’ or is a different labeling scheme used (i.e. just whether an account in the given data is participating in coordination, without respect to the nature of that coordination)?
>
> Answer: The evaluation metrics use the 'normal’ and `coordinated’ labels (i.e., labeled by U.S. Congress investigations of Russian accounts).
>
> (4) Question: How was the prior graph for the 5.2 experiment constructed? Was it the CF version or the TL version?
>
> Answer: For 5.2, we used the CF version (mentioned in implementation details in the Appendix), since it performed better on the unsupervised setting in the IRA experiments.
>
> (5) Comment: For the 5.2 experiment, it may be worth mentioning that the results mirror those observed in other studies on COVID (Horawalavithana Malicious and Low Credibility URLs on Twitter During the AstraZeneca COVID-19 Vaccine Development, 2021, and Cruickshank, Clustering Analysis of Website Usage on Twitter During the COVID-19 Pandemic, 2021) to bolster the results (since there are no labels for validation).
>
> Response: Thank you for your suggestions! We will add more discussions about the above works into our draft.
>
> (6) Comment: The clarity is overall good. The writing is good with a few minor typos (i.e. ‘during during’ on line 231). I do wish there was some kind of overview figure for the method that details how you go from tweet data through the method to coordinated accounts. For Figure 2(b), while I do appreciate the attempt at showing the intuition, I think this figure would be more successful with including intuitive word descriptions of the figures (steps) instead of the mathematical notation.
>
> Response: Thank you for the feedback - we will add an overview figure, with more space given.
>
>
> (6) Comment: The paper is of moderate significance. While it is certainly attacking a very significant problem and displays some positive empirical results there are some issues with its real-world utility. First, the model seems to be predicated upon temporal sequences that include users interacting with tweets, like retweeting. Recent work has shown that other types of coordination, like reposting of the same URLs or reposting the same tweet, but with different mentions (i.e. flooding or spamming) or reporting of the same tweet text from different accounts to also be prevalent and tied to serious disinformation campaigns. In each case, it’s not clear the proposed method would successfully find that coordination.
>
>
> Response: ​​To clarify, the temporal sequences represent the structured input of social media activities, but the model learns the account behaviors in the latent space through account embeddings learned from these observed activities. The embeddings reflect the account's posting patterns, content engagement patterns, interactions with other accounts, and temporal behaviors (from the account ID and timestamp in the temporally ordered sequences).
> Therefore, it captures more complex hidden interactions and coordinated behaviors.
> Moreover, the prior knowledge rules can be customized to represent reposting of the same URLs, same content, co-activity etc. as considered by simply encoding it in the graph (edge-weights). The text features of the tweets can also be easily included as input to the point process model in the proposed framework. Therefore, the proposed framework provides a more convenient, effective and efficient way to represent domain knowledge and guide learning of hidden coordinated behaviors from the observed activities.
>
> (7) Comment: Also, It’s not clear that this method since it relies on interactions with the content and users of one platform, could be extended to multi-platform coordination (for example: Horawalavithana, Twitter Is the Megaphone of Cross-platform Messaging on the White Helmets, 2020, Ng, Multi-platform Information Operations: Twitter, Facebook and YouTube against the White Helmets, 2020, Starbird, Disinformation as Collaborative Work: Surfacing the Participatory Nature of Strategic Information Operations, 2020).
>
> Response: The proposed method does not include any specific platform-specific features. Activity traces of accounts from multiple platforms (temporal sequence of activities) can be collected and the method can be applied. It will still be able to learn characteristic behaviors and hidden interactions of accounts in the embedding space from the activity patterns.

---

### Official Review · Reviewer_hg1U · 2021-07-16

**Rating:** 7
**Confidence:** 3

**Summary:**


In this paper, the author studied the problem of identifying coordinated groups on social media. The authors proposed a detection framework combining neural temporal point process and prior knowledge, and they then designed a theoretically guaranteed variational inference approach to solve the problem of data distribution and sampling. The authors collected a data set and verified the effectiveness of their model and inference on the data set.

**Main Review:**


In general, I think the paper is interesting and the authors make solid contributions. I find the training algorithm of VigDet to be intuitive. Regarding experimental results, I think the selection of the dataset demonstrates the usefulness of the proposed method.

**Time Spent Reviewing:**

4

---

> ### Author Response · Authors · 2021-08-10
> **Response to Reviewer hg1U**
>
> Thank you for the reviews. We sincerely appreciate your feedback on the work.

---

### Official Review · Reviewer_oo8n · 2021-07-27

**Rating:** 7
**Confidence:** 4

**Summary:**

This paper proposes a method for detecting groups of coordinated social media accounts which are engaged in misinformation campaigns.  The approach combines a neural temporal point process model with prior knowledge expressed via temporal logic, fused together using a CRF-style undirected probabilistic model. Training is performed via mean-field VBEM, with an additional lower bound approximation for the M-step.  The model is compared to a number of baselines on a Twitter dataset with Russian bots identified, and is applied to covid misinformation tweets.

**Ethical Concerns:**

No ethical concerns were identified.


**Limitations And Societal Impact:**

This work aims to solve a problem with societal importance, detecting coordinated misinformation campaigns on social media.  Methods for this task could be subject to demographic bias.  It would be worth mentioning that this should be investigated in future work.


**Main Review:**

The task of misinformation coordination detection on social media is a relatively recently formulated machine learning problem, with clear real-world importance.  It is a twist on both community detection and semi-supervised learning on social media with temporal/network/text, involving aspects of each but with some differences as well.  Although the task was not invented in this paper, this work makes a valuable and interesting early contribution to this emerging research direction.

The proposed model, combining neural point process models with graph-based prior knowledge using a joint probability model, is novel in this context, well principled, and potentially useful for solving its intended real-world problem.  There are a couple of missed opportunities in the choices in Section 4.1 to make the model more general and powerful: 1) there is no clear reason why the exponent in equation 4 (the "cubic function") has to be three.  This could be generalized to a selectable hyperparameter. 2) The use of temporal logic to specify the domain knowledge was restricted to very simple rules of a given form (Equation 5).  A bot-creating adversary, knowing that this model was being used, could easily defeat this approach by changing the temporal patterns of coordination. It would be more effective and generalizable if the approach could be extended to domain knowledge from other temporal logic rules.

The VBEM inference approach is reasonable.  It would be better if the implications of Theorem 1 could be formally proved: what are the implications of this lower bound on the convergence of the VBEM algorithm?  I would also like to see more rigor on the claim that the E-step belief propagation "converges at an optimal solution," i.e. prove this result for this specific model, citing specific results in [12] and/or [10] as needed (this could be done in the supplementary materials, for space).

The experimental results are generally solid.  I like that both a real-world problem-relevant dataset (IRA dataset, Twitter with Russian bots) and a larger un-annotated case study dataset (covid misinformation) are used.  A good number of baselines were used on the IRA dataset, and the validation on the covid case study with topic modeling and correlation with various metadata was compelling.  A second ground-truth annotated data would have helped to establish the effectiveness of the method beyond that one dataset.

The paper is generally well written and clear.  There were some minor grammatical errors (see below).

Overall, this is an interesting approach to a fairly new technical problem with real-world importance.  While there is scope for improvement, I am generally favorable towards the work.

Minor comments / typos / grammatical errors:

Line 66: "Our experiments on real world dataset" (add "a")

Line 82: "the performance of prior knowledge based method" (methods plural?)

Line 87: "trys" (tries)

Line 136: "groundtruth" (ground truth)

Line 137,138: "KMeans" ($k$-Means)

Line 143: "Such a method address" (addresses?)

Line 189: "The second term regularize" (regularizes)

Line 218: "Above framework" (add "The")

Line 264: "need threshold" (add "a")

Line 279: "Such phenomenon" (add "a")

Line 311: "suspended accounts is higher" (are higher)

Line 312: "Percentages of recent accounts" (add "The")

-------

Thank you to the authors for the detailed feedback, which helped clarify certain aspects of the work, and reassured me regarding the concerns I mention.  I will stick with my positive rating.

**Time Spent Reviewing:**

3-4 hours

---

> ### Author Response · Authors · 2021-08-10
> **Response to Reviewer oo8n**
>
> Thank you for the reviews! We will correct the typos. Following are our answers to the questions and responses to the comments:
>
> (1) Question: there is no clear reason why the exponent in equation 4 (the "cubic function") has to be three. This could be generalized to a selectable hyperparameter.
>
> Answer: The proposed method can be integrated with any domain-specific prior knowledge rules / functional forms. The exponent of 3 or higher was found to be more beneficial at amplifying differences in the co-occurrence of accounts pairs. Therefore, to make the illustration more clear and simplified, we use 3 in the main content and include the comparison with an exponent as 1 (VigDet-NF) in the appendix. We can update the paper to reflect that the exponent is a selectable hyperparameter in the main content.
>
> (2) Comment: The use of temporal logic to specify the domain knowledge was restricted to very simple rules of a given form (Equation 5). A bot-creating adversary, knowing that this model was being used, could easily defeat this approach by changing the temporal patterns of coordination. It would be more effective and generalizable if the approach could be extended to domain knowledge from other temporal logic rules.
>
> Response: We applied the most general/widely recognized domain knowledge associated with coordinated accounts of co-occurrence in sequences and time-synchronized activities. Using even these general priors with data-driven learning was found to be useful. In practice, we can apply other temporal logic rules r(u,v,S) in Eq.5. The framework is flexible to a broad type of domain-specific priors (that can be represented as a graph).
>
> An adversary might try to change their behavior, but the advantage of the proposed approach is that it is guided by data and domain knowledge, therefore, the signals from the data will still allow the model to discover them. Also, malicious accounts can quickly spread misinformation because they show more definite characteristics of promoting each other’s content, increasing visibility, malicious targeting, and pushing agenda-specific narratives in a timely manner, as opposed to more randomized actions from the normal accounts, which is what we observe in the data. The more the adversary deviates from this behavior pattern, the less effective it will be in spreading misinformation.
>
>
> (3) Comment: The VBEM inference approach is reasonable. It would be better if the implications of Theorem 1 could be formally proved: what are the implications of this lower bound on the convergence of the VBEM algorithm? I would also like to see more rigor on the claim that the E-step belief propagation "converges at an optimal solution," i.e. prove this result for this specific model, citing specific results in [12] and/or [10] as needed (this could be done in the supplementary materials, for space).
>
> Response: For the convergence of the E-step belief propagation, since in the E-step all the parameters are frozen, the updating process is actually a specific case of the belief propagation process in [10]. We will add more formal proof in the supplementary materials.
>
> (4) Comment: The experimental results are generally solid. I like that both a real-world problem-relevant dataset (IRA dataset, Twitter with Russian bots) and a larger un-annotated case study dataset (covid misinformation) are used. A good number of baselines were used on the IRA dataset, and the validation on the covid case study with topic modeling and correlation with various metadata was compelling. A second ground-truth annotated data would have helped to establish the effectiveness of the method beyond that one dataset.
>
> Response: Thank you for your suggestion. We are in the process of investigating and constructing more annotated datasets.
>
> (5) Comment: Methods for this task could be subject to demographic bias. It would be worth mentioning that this should be investigated in future work.
>
> Response: Thank you for the feedback! We will add more detailed discussion in the draft.

---

### Decision · Program_Chairs · 2021-09-28

**Decision:**

Accept (Poster)

**Comment:**

This paper proposes a method for detecting groups of coordinated social media accounts which are engaged in misinformation campaigns based on temporal point processes. All the reviewers recognize that the application is very important as well as challenging and the methodology, even if not groundbreaking, is reasonable for the problem at hand. They found the experimental evaluation solid.

**Consistency Experiment:**

NeurIPS has a long history of experimentation. In 2014, NeurIPS ran an experiment in which 10% of submissions were reviewed by two independent committees to quantify the randomness in the review process. This year, we repeated a variant of this experiment to see how the quality of the review process has changed over time.  This paper was part of the experiment and was therefore assigned to two committees (consisting of reviewers, an Area Chair, and a Senior Area Chair) that reached independent decisions.  If both committees made the same recommendation, this recommendation was followed. If a single committee recommended acceptance, the paper was accepted (with the exception of a few cases in which the other committee identified what we considered a fatal flaw, e.g., an error in a key result).

This copy’s committee reached the following decision: **Accept (Poster)**

The other committee assigned to the paper recommended **Reject**.  You can find the other set of reviews, along with any follow up discussion with the authors here:
https://openreview.net/forum?id=wo_0R04TSrF